# Fast Neural Kernel Embeddings
# for General Activations

**Insu Han**[1]    **Amir Zandieh**[2]    **Jaehoon Lee**[3]
**Roman Novak**[3]    **Lechao Xiao**[3]    **Amin Karbasi**[1,3]

[1]Yale University    [2]Max-Planck-Institut für Informatik    [3]Google Research

## Abstract

Infinite width limit has shed light on generalization and optimization aspects of deep learning by establishing connections between neural networks and kernel methods. Despite their importance, the utility of these kernel methods was limited in large-scale learning settings due to their (super-)quadratic runtime and memory complexities. Moreover, most prior works on neural kernels have focused on the ReLU activation , mainly due to its popularity but also due to the difficulty of computing such kernels for general activations. In this work, we overcome such difficulties by providing methods to work with general activations. First, we compile and expand the list of activation functions admitting exact dual activation expressions to compute neural kernels. When the exact computation is unknown, we present methods to effectively approximate them. We propose a fast sketching method that approximates any multi-layered Neural Network Gaussian Process (NNGP) kernel and Neural Tangent Kernel (NTK) matrices for a wide range of activation functions, going beyond the commonly analyzed ReLU activation. This is done by showing how to approximate the neural kernels using the truncated Hermite expansion of any desired activation functions. While most prior works require data points on the unit sphere, our methods do not suffer from such limitations and are applicable to any dataset of points in $\mathbb{R}^d$. Furthermore, we provide a subspace embedding for NNGP and NTK matrices with near input-sparsity runtime and near-optimal target dimension which applies to any *homogeneous* dual activation functions with rapidly convergent Taylor expansion. Empirically, with respect to exact convolutional NTK (CNTK) computation, our method achieves $106\times$ speedup for approximate CNTK of a 5-layer Myrtle network on CIFAR-10 dataset.

## 1   Introduction

Infinite width limit has enabled fundamental understandings of deep neural networks by establishing a correspondence to kernel methods. In this limit, the network's function prior is a Gaussian process [1–3] and under gradient descent training with squared loss, the network behaves as a linearized function [4, 5]. Underlying these limit, a core object is a neural kernel which encapsulates architectural inductive prior in its functional form [6]. The kernel describing gradient descent dynamics, the Neural Tangent Kernel (NTK) [4], and Neural Network Gaussian Process (NNGP) [2] kernel have been extensively studied [7–12] since they were initially identified. In particular, the infinite width theory has shed light on powerful abilities of deep neural networks including optimization [13–16], generalization [17–19], regularization [20–22] and robustness [23, 24]. Beyond theoretical findings, it has been extensively reported that neural kernels can enhance practical applications including small data classification/regression tasks [25], neural architect search [26, 27], dataset distillation [28, 29], federated learning [30], meta learning [31], generalization attack [32], just to name a few.

Despite those powerful advantages, there is still a gap between practice and theory in the utility of these kernel methods. First, the NNGP and NTK can be exactly computed recursively [2, 4] however, the explicit forms are only known when the corresponding neural networks contain a few set of activation functions such as ReLU or Error functions. While ReLU activation is the default choice for many deep learning applications, recently different activation functions have shown to work well in various domains of machine learning. For example, GeLU [33] has been widely used in Transformer based natural language processing settings [34–36] and sinusoidal activation functions work well for implicit neural representation (e.g. NeRF) [37, 38]. Moreover, Xie et al. [39] showed that smooth activation functions could improve robustness compared to ReLU-based models. To enable better theoretical understanding on the role of these activation functions in these domain, expanding the infinite width limit tool set to general activation function is an important step forward.

Secondly, even if the exact neural kernel computation is explicitly known, it requires significantly huge amount of computing resources. For example, it will take order of few 100 to 1,000 GPU hours to compute the exact NTK of depth 10 convolutional neural networks with pooling on 60,000 CIFAR-10 dataset. High compute requirement is often too expensive to perform extensive studies or use in a practical setting. While Novak et al. [40] have sped up Monte Carlo estimation of the NTK, random sampling remains impractical due to still high kernel computation cost, and cubic (in the training set size) inference cost. Recently, Zandieh et al. [41] proposed an efficient method to approximate the NTK computation via sketching algorithms. Their algorithm can approximate the neural kernels with ReLU activation orders of magnitude faster than the exact one. But it remains unclear how sketching algorithms are extended to other activations.

In this work, we fill this gap by showing that neural kernel for *arbitrary* smooth activation can be expressed in a form of series expansion. We first focus on how to express a kernel function of neural network with a single hidden layer. Under the infinite width limit, this kernel converges to a static function, so-called a *dual kernel*, and is determined by activation in the network. This is a key block to compute the NNGP and NTK of deeper architectures. We establish an explicit expression of dual kernel by expanding activation with the Hermite polynomial basis, and combining it with the fact that Hermite polynomials can play a role of random features of monomial kernels. As a result, our dual kernel formulation relies on coefficients of series expansion of the activation. In addition, we also derive dual kernel expression of the first-order derivative of activation. The NTK can be computed by combining these kernel computations. To the best of our knowledge, our work is the first to study the computation of the NTK for general activations. Furthermore, we provide a subspace embedding for NNGP and NTK matrices with near input-sparsity runtime and near-optimal target dimension. As activation functions play an important role in modern neural network architectures, we hope our work could empower researchers to explore properties of activations in a more principled way. Our main contributions are summarized as follows:

- **Building blocks for infinite-width neural kernel computations**: We derive an explicit expression of the dual kernel for a polynomial activation, which can be a building block for infinite-width neural kernel computations. For non-polynomial activation, we suggest to use its truncated Hermite expansion and analyze an error bound of the dual kernel.

- **Compiling and expanding dual activation** Table 1: We compile various known dual kernel for point-wise activations providing pointers to the original work and expand the set further. We hope our work also serve as an easy reference for various analytic expressions. We emphasize that while many prior references lack required computation for NTK, this work is comprehensive in covering both NNGP/NTK transformations for various activations where analytic computation is possible.

- **NTK computation**: Dual kernels of both activation and its derivative are essential for the NTK computation. Since our formulation requires coefficients of Taylor series of the activation, it is applicable to the dual kernel of derivative of the activation. In addition, we propose how to automatically compute the dual kernel of the derivative without knowing the activation. This approach is useful to characterize the NTK for kernel functions whose activation function is unavailable, e.g., normalized Gaussian, or whose dual kernel of the derivative is unavailable, e.g., GeLU and ELU.

- **Kernel approximation**: We analyze a pointwise error bound of approximated dual kernel via truncated Hermite expansion of the activation with a finite degree. The estimation error can decay polynomially faster in the degree. Furthermore, due to specific decomposition of our kernel formulation, we accelerate the NTK approximation by sketching techniques, similar to [41]. We also propose a new sketching method for the Convolutional NTK with homogeneous activations

Table 1: Activation functions and references for their dual kernels. More detailed expressions are provided in Appendix F.

| Activation | $\sigma(t)$ | Reference for the NNGP | Reference for the NTK |
|---|---|---|---|
| Rectified monomials | $t^q \cdot \mathbb{1}_{\{t \geq 0\}}$ | [44] | [44] |
| Error function | $\text{erf}(t)$ | [43] | [5] |
| ABReLU (Leaky ReLU) | $-A \min(t, 0) + B \max(t, 0)$ | [42, 50, 51] | [42, 50, 51] |
| Exponential | $\exp(At)$ | [46, 52] | [46, 52] |
| Hermite polynomials | $h_q(t)$ | [46] | This work |
| Sinusoidal | $\sin(At + B)$ | [45, 47, 53] | This work |
| Gaussian | $\exp\left(-At^2\right)$ | [43] | This work |
| GeLU | $\frac{t}{2}\left(1 + \text{erf}\left(\frac{t}{\sqrt{2}}\right)\right)$ | [48] | This work |
| ELU | $\text{step}(t)t + \text{step}(-t)\left(e^t - 1\right)$ | [48] | This work |
| Normalized Gaussian | Unknown | [54] | This work |
| RBF | $\sqrt{2}\sin(\sqrt{2A}t + \frac{\pi}{4})$ | [45] | This work |
| Gabor | $\exp(-t^2)\sin(t)$ | This work | This work |
| Monomial | $t^q$ | This work | This work |
| Polynomial | $\sum_{j=0}^{q} a_j t^j$ | This work | This work |

and analyze both a pointwise error bound and its runtime in Appendix D.2. Notably, our sketching method's runtime scales only linearly in the number of pixels of the input images, while the exact CNTK computation scales quadratically in the number of pixels.

• **Implementation**: We open-source NNGP and NTK for new activations within the Neural Tangents library [42] and sketching algorithm at `https://github.com/insuhan/ntk_activations`.

## 1.1 Related Work

Neural kernels (NTK, NNGP) can be computed using the recursive formula [2–5]. A prerequisite for these kernels is computing a static kernel function which is defined as the expectation of some function of (non-linear) activation in neural network over the standard normal distribution. Williams [43] studied this a dual kernel of $\text{erf}(t)$ and Gaussian. Cho and Saul [44] derived dual kernels for the rectified monomials, i.e., $t^q \mathbb{1}_{\{t \geq 0\}}$, this function is equal to arc-cosine kernels where ReLU activation is a special case when $q = 1$. Rahimi and Recht [45] showed that sinusoidal activations, e.g., $\sin$ or $\cos$, can result in the Gaussian RBF kernel function using the Fourier transform. Daniely et al. [46] proposed a method to obtain a dual kernel if activation can be expanded by Hermite polynomials. However, inputs of the resulting kernels are restricted to be on the unit sphere. Louart et al. [47] analyzed asymptotic properties of dual kernel with random matrix theory and show closed-form formula of such as $\text{erf}$, $|t|$, sinusoidal. Tsuchida et al. [48] studied the dual kernels of both Gaussian Error Linear Unit (GeLU) [33] and Exponential Linear Unit (ELU) [33]. For activation that does not admit a closed-form expression, Lee et al. [2] numerically computed dual activation by doing interpolation on predetermined grid of variances and covariances. Table 1 summarizes activations whose dual kernels were priorly known, as well as expanding (in this work) the set to previously unknown expressions. Recently, Simon et al. [49] discovered that NTK of fully-connected neural network with any depth can be converted into that of a 1 hidden-layer neural network by modifying activation function. However, their method is limited to the normalized input data and fully-connected networks.

## 2 Preliminaries

**Notations.** We denote the identity matrix of dimension $d$ by $\boldsymbol{I}_d$. For a scalar function $f$, we write $f^{(k)}$ to denote its $k$-th derivative. We use $\mathbb{1}_{\mathcal{E}}$ to denote the indicator of event $\mathcal{E}$. For a smooth function $\sigma : \mathbb{R} \to \mathbb{R}$, we use $\sigma^{(k)}$ to denote its $k$-th derivative and define $\|\sigma\|_{\mathcal{N}(0,\nu^2)}^2 := \mathbb{E}_{t \sim \mathcal{N}(0,\nu^2)}[|\sigma(t)|^2]$ for some $\nu \in \mathbb{R}$ and simply write $\|\sigma\|_{\mathcal{N}(0,1)} := \|\sigma\|_{\mathcal{N}}$. For scalar functions $f, g$ we use $f \circ g$ to denote the composition of these functions and $f^{\circ q}$ to denote the $q$ times self-composition of $f$, e.g., $f^{\circ 3}(x) = f(f(f(x)))$. Given a positive semidefinite matrix $\boldsymbol{K}$ and $\lambda > 0$, the statistical dimension of $\boldsymbol{K}$ with regularizer $\lambda$ is defined as $s_\lambda(\boldsymbol{K}) := \text{tr}(\boldsymbol{K}(\boldsymbol{K} + \lambda \boldsymbol{I})^{-1})$. We use $\text{nnz}(x)$ to denote the number of nonzero entries in $x$. Given $x \in \mathbb{R}^m$ and $y \in \mathbb{R}^n$, we define $x \otimes y :=$

$[x_1y_1, x_2y_1, \ldots x_my_1, x_1y_2, \ldots x_my_2, \ldots x_my_n]$ and $x^{\otimes p}$ as the $p$-fold self-tensoring of $x$. We also define $\oplus$ as the direct sum between vectors.

**Hermite polynomials.** The *Probabilist's Hermite polynomials* of degree $\ell \geq 0$ is defined as

$$h_\ell(t) = (-1)^\ell e^{\frac{t^2}{2}} \left[ \frac{d^\ell}{dt^\ell} e^{-\frac{t^2}{2}} \right] = \ell! \sum_{i=0}^{\lfloor \ell/2 \rfloor} \frac{(-1)^i}{i!(\ell - 2i)!} \frac{t^{\ell - 2i}}{2^i}. \tag{1}$$

The polynomials $\{h_\ell\}_{\ell \geq 0}$ form a set of orthogonal basis for the space of square-integrable functions in $\mathbb{R}$ with respect to the normal measure $\mathcal{N}(0,1)$, i.e., the $L^2$ space of functions $L^2(\mathbb{R}, \mathcal{N}) := \{f : \mathbb{R} \to \mathbb{R} \mid \|\sigma\|_{\mathcal{N}}^2 < \infty\}$. Particularly, it holds that $\mathbb{E}_{t \sim \mathcal{N}(0,1)}[h_\ell(t) h_m(t)] = \ell! \cdot \mathbb{1}_{\{\ell = m\}}$. Thus, any function $f \in L^2(\mathbb{R}, \mathcal{N})$ has a unique Hermite expansion in the sense of $\|f - \sum_{t=0}^{\infty} c_j h_j\|_{\mathcal{N}} = 0$ and coefficient $c_j$ can be computed as $c_j = \mathbb{E}_{t \sim \mathcal{N}(0,1)}[f(t) h_j(t)] / j!$.

**Infinite width neural kernels.** Given an activation $\sigma : \mathbb{R} \to \mathbb{R}$ satisfying that $\|\sigma\|_{\mathcal{N}} = 1$, consider a fully-connected $L$-layered neural network $f : \mathbb{R}^d \to \mathbb{R}$ for $L \geq 2$ defined as[1]

$$f_\sigma(x; \mathcal{W}) = \left\langle w^{(L)}, z_{L-1} \right\rangle / \sqrt{d_{L-1}}, \;\; z_\ell = \sigma\left( \boldsymbol{W}^{(\ell)} z_{\ell-1} / \sqrt{d_{l-1}} \right), \;\; z_0 = x \tag{2}$$

where $\mathcal{W} := \text{vec}\left( w^{(L)}, \cup_{\ell=1}^{L-1} \boldsymbol{W}^{(\ell)} \right)$ for $w^{(L)} \in \mathbb{R}^{d_{L-1}}, \boldsymbol{W}^{(\ell)} \in \mathbb{R}^{d_\ell \times d_{\ell-1}}, d_0 := d, d_l := m$ for $l > 0$ is a collection of learnable parameters, $m$ is the width of the network, and $\sigma(\cdot)$ is applied point-wisely. In the infinite width limit, i.e., $m \to \infty$, when all elements of $\mathcal{W}$ are initialized by i.i.d. random samples from $\mathcal{N}(0,1)$ and optimized via gradient descent on the least-square loss with an infinitesimal learning rate, the prediction of trained network becomes identical to that of its first order Taylor approximation at $\mathcal{W}$. Hence, inference with such ultra-wide network is equivalent to kernel regression with a static kernel, the so-called Neural Tangent Kernel (NTK), defined as $\Theta_\sigma^{(L)}(x, y) := \text{plim}_{m \to \infty} \langle \nabla_{\mathcal{W}} f_\sigma(x; \mathcal{W}), \nabla_{\mathcal{W}} f_\sigma(y; \mathcal{W}) \rangle$ (convergence in probability to a constant). In addition, at initialization the output of an infinitely wide network is equivalent to a sample from a Gaussian process with mean zero and covariance $\Sigma_\sigma^{(L)}(x, y) := \text{plim}_{m \to \infty} \langle f_\sigma(x; \mathcal{W}), f_\sigma(y; \mathcal{W}) \rangle$, known as the Neural Network Gaussian Process (NNGP) kernel.

**Recursive expression for NNGP and NTK.** Several previous works [2–5] have shown that the NNGP and NTK can be expressed using the following recursive procedure:

1. For every $x, y \in \mathbb{R}^d$, let $K_\sigma^{(0)}(x, y) := \langle x, y \rangle$ and for every layer $h = 1, \ldots, L$, recursively define kernel functions $K_\sigma^{(h)}, \dot{K}_\sigma^{(h)} : \mathbb{R}^d \times \mathbb{R}^d \to \mathbb{R}$ as:

$$K_\sigma^{(h)}(x, y) := \mathbb{E}_{(u,v) \sim \mathcal{N}(0, \boldsymbol{\Lambda}_\sigma^{(h)})} [\sigma(u)\sigma(v)], \quad \dot{K}_\sigma^{(h)}(x, y) := \mathbb{E}_{(u,v) \sim \mathcal{N}(0, \boldsymbol{\Lambda}_\sigma^{(h)})} [\sigma'(u)\sigma'(v)], \tag{3}$$

where the covariance matrix is $\boldsymbol{\Lambda}_\sigma^{(h)} := \begin{bmatrix} K_\sigma^{(h-1)}(x, x) & K_\sigma^{(h-1)}(x, y) \\ K_\sigma^{(h-1)}(y, x) & K_\sigma^{(h-1)}(y, y) \end{bmatrix} \in \mathbb{R}^{2 \times 2}$.

2. The depth-$L$ NNGP kernel is $K_\sigma^{(L)}(x, y)$ and the depth-$L$ NTK $\Theta_\sigma^{(L)}$ can be recursively computed as $\Theta_\sigma^{(0)}(x, y) := \langle x, y \rangle$ and

$$\Theta_\sigma^{(h)}(x, y) := \Theta_\sigma^{(h-1)}(x, y) \cdot \dot{K}_\sigma^{(h)}(x, y) + K_\sigma^{(h)}(x, y). \tag{4}$$

At the core of the expression for $\Theta_\sigma^{(L)}$, there is the expectation term over 2-dimensional Gaussian distribution in Equation (3). This expectation term for the case where both diagonal entries of the covariance matrix $\boldsymbol{\Lambda}_\sigma^{(\ell)}$ are equal to one, was previously studied in [46]. We extend this to encompass general symmetric covariance matrices in the following definition.

---

[1]Throughout the paper, we consider scalar-valued networks without biases for simplicity, but this can be extended to vector-valued networks with biases . We also assume $\|\sigma\|_{\mathcal{N}} = 1$ which does not change our results.

**Definition 1** (Dual Activation and Dual Kernel). For a smooth $\sigma : \mathbb{R} \to \mathbb{R}$, we define the *Dual Kernel* of $\sigma$ as $K_\sigma : \mathbb{R}^d \times \mathbb{R}^d \to \mathbb{R}$ defined as

$$K_\sigma(x, y) := \mathbb{E}_{w \sim \mathcal{N}(0, I_d)} [\sigma(\langle w, x \rangle) \sigma(\langle w, y \rangle)] \quad \text{for every } x, y \in \mathbb{R}^d. \tag{5}$$

Equation (5) only depends on bivariate Gaussian random variables $\langle w, x \rangle, \langle w, y \rangle$ where $\mathbb{E}[\langle w, x \rangle^2] = \|x\|_2^2, \mathbb{E}[\langle w, y \rangle^2] = \|y\|_2^2$ and $\mathbb{E}[\langle w, x \rangle \cdot \langle w, y \rangle] = \langle x, y \rangle$. Hence one can look at the dual kernel from a different perspective by choosing a proper covariance matrix. To this end, let $\mathbf{\Lambda}_{a,b,c} := \begin{bmatrix} a^2 & abc \\ abc & b^2 \end{bmatrix}$ for every $a, b \in \mathbb{R}_+$ and $c \in [-1, 1]$ and the *Dual Activation* of $\sigma$ with respect to $\mathbf{\Lambda}_{a,b,c}$ is the function $k_\sigma : \mathbb{R}_+ \times \mathbb{R}_+ \times [-1, 1] \to \mathbb{R}$ defined as $k_\sigma(a, b, c) := \mathbb{E}_{(u,v) \sim \mathcal{N}(0, \mathbf{\Lambda}_{a,b,c})} [\sigma(u)\sigma(v)]$.

With these definitions in place, the following relationship between dual kernel and activation holds

$$K_\sigma(x, y) = k_\sigma \left( \|x\|_2, \|y\|_2, \frac{\langle x, y \rangle}{\|x\|_2 \|y\|_2} \right). \tag{6}$$

Observe that $K_\sigma(x, y)$ corresponds to the NNGP kernel of a 1-hidden layer neural network with activation $\sigma$. For some specific activations, e.g., ReLU, Error function, closed form expressions for their dual activations are known (see Table 1). Hence, one can compute the NTK analytically when dual kernels of the activation and its derivative have a closed form expression. The above also holds for kernels corresponding to convolutional neural networks called CNN-GP [7, 8] and CNTK [9].

## 3 NNGP and NTK for Smooth Activations

In this section, we focus on the NNGP and NTK for a wide range of smooth activation functions. We first show that a series expansion for the dual kernel can be obtained from that of the activation function, which is a key to NNGP kernel computation. By applying this result to the derivative of the activation function, we can also compute the NTK for the same activation.

### 3.1 Dual Kernel Computation

Daniely et al. [46] proved that for absolutely continuous $\sigma : \mathbb{R} \to \mathbb{R}$ and any $x, y \in \mathbb{S}^{d-1}$, the dual kernel is equal to $K_\sigma(x, y) = \sum_{j=0}^\infty c_j^2 \, j! \cdot \langle x, y \rangle^j$ where $\{c_j\}_{j \geq 0}$ are coefficients of Hermite expansion of $\sigma$. We now proceed to generalize this result from $\mathbb{S}^{d-1}$ to entire $\mathbb{R}^d \setminus \{0\}$. First we remark that it can be naturally extended to the dual kernel of *q-homogeneous* activation functions, i.e., $\sigma(at) = |a|^q \sigma(t)$ for every $a, t \in \mathbb{R}$, on the entire $\mathbb{R}^d \setminus \{0\}$. For every $x, y \in \mathbb{R}^d \setminus \{0\}$, the corresponding dual kernel is

$$K_\sigma(x, y) = \|x\|_2^q \|y\|_2^q \cdot \sum_{j=0}^\infty c_j^2 \, j! \cdot \left( \frac{\langle x, y \rangle}{\|x\|_2 \|y\|_2} \right)^j. \tag{7}$$

As examples, (leaky) ReLU and rectified polynomials fall into this activation class.

Now suppose that $\sigma$ is not homogeneous. In particular, we first consider a polynomial activation $\sigma(t) = \sum_{j=0}^q a_j t^j$ with coefficients $\{a_j\}_{j=0}^q$. Recall that $K_\sigma(x, y)$ can be obtained by taking the expectation of $\sigma(\langle w, x \rangle) \sigma(\langle w, y \rangle)$ over $w \sim \mathcal{N}(0, I_d)$ for every $x, y \in \mathbb{R}^d \setminus \{0\}$. To make use of Daniely et al. [46]'s result, we factorize the input into its radial and angular part and rewrite the activation by expressing monomials in the Hermite polynomial basis. Formally, let us write monomials in the Hermite basis as $t^i = \sum_{\ell=0}^i \mu_{i,\ell} h_\ell(t)$ for some coefficients $\{\mu_{j,i}\}_{i=0}^j$. Then

$$\sigma(\langle w, x \rangle) = \sum_{j=0}^q a_j \|x\|_2^j \left\langle w, \frac{x}{\|x\|_2} \right\rangle^j = \sum_{i=0}^q \left( \sum_{j=i}^q \mu_{j,i} \|x\|_2^j a_j \right) h_i \left( \left\langle w, \frac{x}{\|x\|_2} \right\rangle \right). \tag{8}$$

Then, we can derive the dual kernel of polynomial activation. We further relax a condition on the activation and propose the result below.

**Theorem 1.** *For a polynomial $\widetilde{\sigma}(t) = \sum_{j=0}^{q} a_j t^j$, the dual kernel of $\widetilde{\sigma}(\cdot)$, as per Definition 1, is*

$$K_{\widetilde{\sigma}}(x,y) := \sum_{\ell=0}^{q} r_{\widetilde{\sigma},\ell}(\|x\|_2) \, r_{\widetilde{\sigma},\ell}(\|y\|_2) \left( \frac{\langle x, y \rangle}{\|x\|_2 \|y\|_2} \right)^{\ell} \tag{9}$$

*where $r_{\widetilde{\sigma},\ell}(t) := \sum_{i=0}^{\lfloor \frac{q-\ell}{2} \rfloor} \frac{a_{\ell+2i}(\ell+2i)!}{2^i \cdot i! \cdot \sqrt{\ell!}} t^{2i+\ell}$. Moreover, if an activation function $\sigma : \mathbb{R} \to \mathbb{R}$ satisfies $\|\sigma\|^2_{\mathcal{N}(0,\nu^2)} < \infty$ and $\|\sigma - \widetilde{\sigma}\|^2_{\mathcal{N}(0,\nu^2)} \leq \varepsilon$ for some $\varepsilon > 0$ and $\nu \geq 1$, then for every $x, y \in \mathbb{R}^d$ such that $\|x\|_2, \|y\|_2 \in (0, \nu]$ the following holds*

$$|K_{\sigma}(x,y) - K_{\widetilde{\sigma}}(x,y)| \leq \sqrt{\frac{\nu^2 \cdot \varepsilon \left( 6\|\sigma\|^2_{\mathcal{N}(0,\nu^2)} + 4\varepsilon \right)}{\|x\|_2 \|y\|_2}}. \tag{10}$$

The proof of Theorem 1 is provided in Appendix B.2. For non-polynomial activations, one can consider approximating $\sigma$ with its Hermite or Taylor expansion and then apply Theorem 1. Examples can be found in Appendix B.2. For activation functions that do not have a Taylor expansion but are $k$-th order differentiable, we show that, using their Hermite expansion, one can obtain a good approximation to the corresponding dual kernel.

**Theorem 2.** *Given $\sigma : \mathbb{R} \to \mathbb{R}$, suppose that there exists an integer $k \geq 2$ and some $\nu \geq 1$ such that for every $i = 0, \ldots, k$, $\sigma^{(i)}$ is absolutely continuous and $\lim_{t \to \pm\infty} e^{-\frac{t^2}{4}} \sigma^{(i)}(\nu t) = 0$ and moreover $\|\sigma\|^2_{\mathcal{N}(0,\nu^2)} < \infty$ and $\|\sigma^{(k)}\|^2_{\mathcal{N}(0,\nu^2)} < \infty$. Consider the Hermite expansion coefficients $\{c_j\}_{j\geq 0}$ of function $\sigma(\nu t)$ and denote $\widetilde{\sigma}(t) := \sum_{j=0}^{q} c_j h_j(t/\nu)$. Given $x, y \in \mathbb{R}^d$ with $\|x\|_2, \|y\|_2 \in (0, \nu]$,*

$$|K_{\sigma}(x,y) - K_{\widetilde{\sigma}}(x,y)| \leq \frac{5\nu^{k+1} \|\sigma^{(k)}\|_{\mathcal{N}(0,\nu^2)} \max\left( \|\sigma\|_{\mathcal{N}(0,\nu^2)}, \nu^k \|\sigma^{(k)}\|_{\mathcal{N}(0,\nu^2)} \right)}{\sqrt{\|x\|_2 \|y\|_2 \cdot k \cdot q^{k-1}}}. \tag{11}$$

*where $K_{\sigma}(\cdot, \cdot)$ and $K_{\widetilde{\sigma}}(\cdot, \cdot)$ are dual kernels corresponding to $\sigma(\cdot)$ and $\widetilde{\sigma}(\cdot)$ in Definition 1, respectively. Moreover, for the ReLU activation $\sigma(t) = \max(t, 0)$, it holds that*

$$|K_{\sigma}(x,y) - K_{\widetilde{\sigma}}(x,y)| \leq \sqrt{\frac{2\nu^6}{q\|x\|_2 \|y\|_2}}. \tag{12}$$

The proof of Theorem 2 is provided in Appendix B.3. Observe that when the activation is $k$-th order differentiable and the norms of its derivative and inputs are bounded then the approximation error decreases with $\mathcal{O}(\frac{1}{\sqrt{kq^{k-1}}})$ rate. In Section 5, we empirically evaluate the dual kernel of various activations using Hermite expansion and verify that smooth activations (e.g., Gaussian or sinusoidal) provides much lower approximation errors than non-smooth ones (e.g., ReLU).

## 3.2 NNGP and NTK Computations

Once dual kernels of $\sigma$ and $\sigma'$ or their polynomial approximations are calculated, one can compute (approximate) NNGP and NTK using Theorem 1 or Theorem 2 and the recursion in Equation (4). However, there are scenarios where we are only given the dual kernel and the corresponding activation or derivative of the activation is unknown to us. For example, Shankar et al. [54] devised a normalized Gaussian kernel defined as

$$K_G(x,y) = \|x\|_2 \|y\|_2 \exp\left( \frac{\langle x, y \rangle}{\|x\|_2 \|y\|_2} - 1 \right), \tag{13}$$

and reported that NNGP with this dual kernel performs better than the ReLU NTK by showing promising results on various tasks. Note that, recovering the activation from $K_G$ is non-trivial. From the dual kernel perspective, the activation should be 1-homogeneous and its Hermite series expansion is of form $\sum_{j=0}^{\infty} \frac{\pm 1}{j!} h_j(t)$ and it is generally unknown how to choose the sign pattern on coefficients of this series that would satisfy homogeneity constraint. Instead of trying to recover the activation from dual kernel, we show how to directly derive the dual kernel of derivative of activation without knowing the activation.

**Theorem 3.** *Given a differentiable activation function $\sigma : \mathbb{R} \to \mathbb{R}$ which satisfies $|\sigma(t)| \le C_1 \exp\left(\frac{t^2}{4.1\nu^2}\right)$, $|\sigma'(t)| \le C_2 \exp\left(\frac{t^2}{4.1\nu^2}\right)$, $\|\sigma\|^2_{\mathcal{N}(0,\nu^2)} < \infty$ and $\|\sigma''\|^2_{\mathcal{N}(0,\nu^2)} < \infty$ for some $\nu \ge 1$ and constants $C_1, C_2$, the following holds for any $x, y \in \mathbb{R}^d$ with $\|x\|_2, \|y\|_2 \in (0, \nu]$ and $|\langle x, y \rangle| < \|x\|_2 \|y\|_2$:*

$$K_{\sigma'}(x,y) = \frac{1}{\|x\|_2 \|y\|_2} \left. \frac{\partial}{\partial c} k_\sigma\left(\|x\|_2, \|y\|_2, c\right) \right|_{c = \frac{\langle x,y \rangle}{\|x\|_2 \|y\|_2}}. \tag{14}$$

*Additionally, if $\frac{\partial}{\partial c} k_\sigma(\cdot, \cdot, c)$ is continuous at $c = \pm 1$ then Equation (14) holds for $x, y$ such that $|\langle x, y \rangle| = \|x\|_2 \|y\|_2$.*

The proof of Theorem 3 is provided in Appendix B.5. Our result is more general compared to [49] where the previous work assumes that the Hermite expansion of given activation should converge and $\|x\|_2 = \|y\|_2$. Applying Theorem 3 to Equation (13) provides that $\dot{K}_G(x,y) = \exp\left(\frac{\langle x,y \rangle}{\|x\|_2 \|y\|_2} - 1\right)$ hence one can compute the NTK function even if the corresponding activation is unknown. In the previous work [54], only "NNGP" performances of the normalized Gaussian kernel were reported.

Moreover, with Theorem 3, only the knowledge of dual activation suffices to compute both NNGP and NTK. For example, while dual activation (thus NNGP) of GeLU was known in Tsuchida et al. [48], $k_{\sigma'}$ was not derived explicitly. Theorem 3 provides a simple way to compute $k_{\sigma'}$ (given in Equation (126)) via automatic differentiation, without requiring to take the expectation under multivariate Gaussian distribution or computing derivatives by hand. This is implemented in `stax_extensions.Elementwise` in our code supplement. Our method allows to omit the enitre effort, lines of code, and potential mistakes in deriving and implementing the NTK.

### 3.3 Gauss-Hermite Quadrature

One simple approach to obtain dual activation function for general activation functions without closed form expressions is to evaluate the expectation of under the $2d$ Gaussian distribution as numerical integration. This can be efficiently done by Gauss-Hermite quadrature

$$k_\sigma(a,b,c) \approx \frac{1}{\pi} \sum_{i=1}^{q} \sum_{j=1}^{q} w_i w_j \left[ \sigma(\sqrt{2}ax_i) \cdot \sigma(\sqrt{2}bcx_i + \sqrt{2}b\sqrt{1-c^2}x_j) \right] \tag{15}$$

where $(x_i, w_i)$, correspond to $i$-th root of degree $q$ Hermite polynomial $h_i(x)$ and associated weights [55] $w_i = \frac{q!\sqrt{\pi}}{q^2(h_{q-1}(\sqrt{2}x_i))^2}$. See Appendix E for the derivation of the quadrature formula.

For smooth activation functions errors will quickly go down as $q$ increases by Theorem 2. We use this method to compute approximate (non-sketched) kernels for general activation functions in Figure 3 and Figure 4. It is implemented as `stax_extensions.ElementwiseNumerical` in our code.

## 4 Approximating Neural Kernels via Sketching

Although using our Theorem 1, Theorem 2, and Theorem 3, one can analytically compute NTK for general activation functions, computing all entries in the NTK kernel matrix requires massive amount of resources, i.e., $\Omega(n^2(d + Lq^2))$ runtime and $\Omega(n^2)$ memory for datasets with $n$ points in $\mathbb{R}^d$. This becomes even more expensive for CNTK, where its runtime can be $\Omega((nd_1d_2)^2(c + Lq^2))^2$ for $n$ of images with size $d_1 \times d_2 \times c$. To avoid quadratic complexities, we adopt a fast and efficient feature map construction via randomized sketching [41] for both NTK and NNGP, i.e.,

$$\Theta_\sigma^{(L)}(x,y) \approx \left\langle \psi^{(L)}(x), \psi^{(L)}(y) \right\rangle, \quad K_\sigma^{(L)}(x,y) \approx \left\langle \phi^{(L)}(x), \phi^{(L)}(y) \right\rangle. \tag{16}$$

The previous approach was only applicable for the ReLU activation but we establish more general scheme based on our new results for dual kernel approximation.

---

[2]This is assuming Hermite expansion degree $q$, when exact expression is known $q^2$ is constant.

---

**Algorithm 1** Subspace Embedding of Homogeneous NNGP and NTK

---

1: **input**: $x \in \mathbb{R}^d$, depth $L$, sketching dimension $m$, polynomial $\widetilde{\kappa}(t) = \sum_{j=0}^q a_j t^j$ with $a_j \in \mathbb{R}_+$

2: calculate the polynomial $P^{(L)}(t) = \widetilde{\kappa}^{\circ L}(t) = \sum_{j=0}^{q^L} b_j t^j$ with coefficients $b_j \in \mathbb{R}_+$

3: calculate the polynomial $R^{(L)}(t) = \sum_{h=0}^L \widetilde{\kappa}^{\circ h}(t) \cdot \prod_{i=h}^{L-1} \widetilde{\kappa}' \circ \widetilde{\kappa}^{\circ i}(t) = \sum_{j=0}^p c_j t^j$ with coefficients $c_j \in \mathbb{R}_+$ and degree $p = q^{\mathcal{O}(L)}$

4: for $\ell = 0, \dots, p$, let $Q^\ell \in \mathbb{R}^{m \times d^\ell}$ be a degree-$\ell$ POLYSKETCH (See Appendix A)

5: for every $\ell = 0, \dots, p$, $u^\ell \leftarrow Q^\ell \left( \frac{x}{\|x\|_2} \right)^{\otimes \ell}$

6: construct $\phi^{(L)}(x) \leftarrow \|x\|_2 \cdot \bigoplus_{j=0}^{q^L} \sqrt{b_j} u^j$ and $\psi^{(L)}(x) \leftarrow \|x\|_2 \cdot \bigoplus_{j=0}^p \sqrt{c_j} u^j$

7: **return** $\phi^{(L)}(x)$ (NNGP embedding), $\psi^{(L)}(x)$ (NTK embedding)

---

**Subspace embedding for homogeneous dual kernels.** We provide a subspace embedding for NNGP and NTK matrices with near input-sparsity runtime and near-optimal target dimension which applies to any *homogeneous* dual activation functions with rapidly convergent Taylor expansion. More specifically, we call a dual kernel $K_\sigma$ homogeneous if there exists a positive definite dot-product kernel function $\kappa : [-1, 1] \to [-1, 1]$ such that,

$$K_\sigma(x, y) = \|x\|_2 \|y\|_2 \cdot \kappa \left( \frac{\langle x, y \rangle}{\|x\|_2 \|y\|_2} \right). \tag{17}$$

For such homogeneous dual kernels, the NTK and NNGP take a similar homogeneous form. In fact, one can show by induction that when the dual kernel is in form of Equation (17), the depth-$L$ NNGP function defined in Equation (3) is equal to the following for any positive integer $L$,

$$K_\sigma^{(L)}(x, y) = \|x\|_2 \|y\|_2 \cdot \kappa^{\circ L} \left( \frac{\langle x, y \rangle}{\|x\|_2 \|y\|_2} \right), \tag{18}$$

where $\kappa^{\circ L}$ denoted the $L$-fold composition of function $\kappa$. Furthermore, if $\kappa$ has a derivative $\kappa' : [-1, 1] \to [-1, 1]$, using Theorem 3, there exists a depth-$L$ NTK for this dual kernel, equal to

$$\Theta_\sigma^{(L)}(x, y) = \|x\|_2 \|y\|_2 \cdot \left. \sum_{h=0}^L \kappa^{\circ h}(t) \cdot \prod_{i=h}^{L-1} \kappa' \circ \kappa^{\circ i}(t) \right|_{t = \frac{\langle x, y \rangle}{\|x\|_2 \|y\|_2}}, \tag{19}$$

where we use the convention that $\kappa^{\circ 0}(t) = t$. Therefore, if $\kappa(\cdot)$ can be tightly approximated by a low-degree polynomial, then the NNGP and NTK functions can also be tightly approximated by low-degree polynomials. Thus, by applying POLYSKETCH, which is a norm-preserving dimensionality reduction that can be applied to the tensor product of multiple vectors very quickly [56], to the polynomial approximations to these kernels, we can spectrally approximate the NNGP and NTK kernel matrices. For details on POLYSKETCH see Appendix A. We provide the details of this procedure in Algorithm 1 and prove the correctness and runtime of our procedure in Theorem 4.

**Theorem 4** (Homogeneous NTK Embedding). *Suppose that the dual kernel $K_\sigma$ is homogeneous as per Equation (17). Also suppose $\widetilde{\kappa}(t)$ is a degree-$q$ polynomial with non-negative coefficients that satisfies (**1**) $\max_{t \in [-1, 1]} |\widetilde{\kappa}(t) - \kappa(t)| \leq \frac{1}{\text{poly}(n)}$ and $\max_{t \in [-1, 1]} |\widetilde{\kappa}'(t) - \kappa'(t)| \leq \frac{1}{\text{poly}(n)}$, (**2**) $\max_{|t| \leq 1 + \frac{1}{\text{poly}(n)}} |\widetilde{\kappa}(t + \gamma) - \widetilde{\kappa}(t)| \leq \frac{1}{\text{poly}(n)}$ and $\max_{|t| \leq 1 + \frac{1}{\text{poly}(n)}} |\widetilde{\kappa}'(t + \gamma) - \widetilde{\kappa}'(t)| \leq \frac{1}{\text{poly}(n)}$ for any $|\gamma| \leq \frac{1}{\text{poly}(n)}$. Then for any integer $L \geq 1$, any $\varepsilon, \lambda \geq \frac{1}{\text{poly}(n)}$, and any dataset $X \in \mathbb{R}^{d \times n}$ with $\|X\|_F \leq \text{poly}(n)$, if $K_{\text{ntk}} \in \mathbb{R}^{n \times n}$ is the depth-$L$ NTK kernel matrix on this dataset, there exists $m = \mathcal{O} \left( \frac{s_\lambda(K_{\text{ntk}})}{\varepsilon^2} \cdot \text{poly}(q^L, \log n) \right)$ such that the output $\psi^{(L)}(X) \in \mathbb{R}^{m \times n}$ of Algorithm 1 satisfies with probability at least $1 - \frac{1}{\text{poly}(n)}$*

$$(1 - \varepsilon) (K_{\text{ntk}} + \lambda I_n) \preceq \psi^{(L)}(X)^\top \psi^{(L)}(X) + \lambda I_n \preceq (1 + \varepsilon) (K_{\text{ntk}} + \lambda I_n). \tag{20}$$

*Moreover, the runtime of Algorithm 1 is $\mathcal{O} \left( \text{poly}(q^L, \log n) \cdot \varepsilon^{-2} \cdot (s_\lambda(K_{\text{ntk}}) \cdot n + \text{nnz}(X)) \right)$.*

We prove this theorem in Appendix C. As an example, let us apply Theorem 4 on the normalized Gaussian kernel $K_G$ defined in Equation (13), which is homogeneous. The dot-product factor corresponding to this dual kernel is $\kappa(t) = \exp(t - 1)$. The truncated Taylor series of this

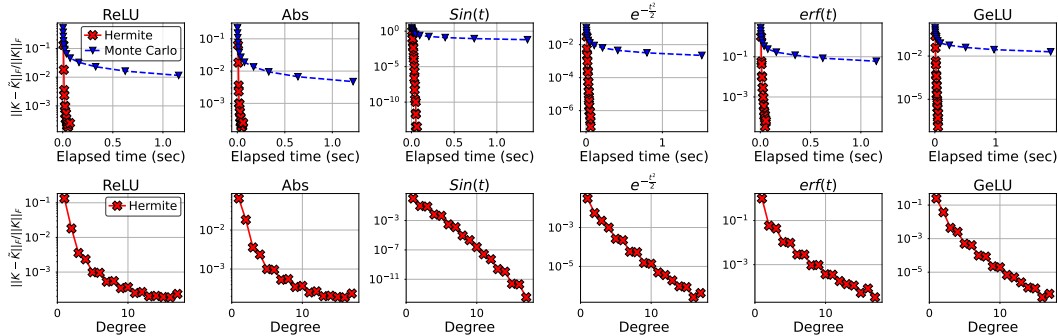

Figure 1: Relative errors of dual kernel approximations via the truncated Hermite expansion and Monte Carlo estimation under synthetic dataset with $n = 1,000, d = 256$.

function is $\widetilde{\kappa}(t) = \sum_{j=0}^{q} \frac{t^j}{e \cdot j!}$. If $q = \Omega(\log n)$ then it can be verified that the polynomial $\widetilde{\kappa}(t)$ satisfies the preconditions of Theorem 4. Therefore, one can invoke Algorithm 1 to get a subspace embedding for the NTK kernel matrix corresponding to the normalized Gaussian dual kernel $K_G$ in $\mathcal{O}\left(\varepsilon^{-2} \cdot (s_\lambda(\boldsymbol{K}_{\mathrm{ntk}}) \cdot n + \mathrm{nnz}(\boldsymbol{X})) \cdot \mathrm{poly}\left(\log^L n\right)\right)$ time and with a target dimension of $m = \mathcal{O}\left(\varepsilon^{-2} \cdot s_\lambda(\boldsymbol{K}_{\mathrm{ntk}}) \cdot \mathrm{poly}\left(\log^L n\right)\right)$. For any constant number of layers, $L$, this runtime and target dimension is is optimal up to $\mathrm{poly}(\log n)$ factors. The implementation of our sketching algorithm is available at https://github.com/insuhan/ntk_activations.

## 5 Experiments

In this section, we perform experiments with the proposed neural kernels based on our dual kernel approximation. All experiments run using a single A100 GPU machine.

**Kernel approximation.** We first benchmark our algorithm to approximate the dual kernel matrix. We use ReLU, Abs (i.e., $\sigma(t) = |t|$), sin, Gaussian, erf and GeLU activations and approximate them by their Hermite expansion where degree changes from $q = 1$ to 20. We randomly generate $n = 1,000$ of 256-dimensional inputs where each entry is i.i.d. drawn from $\mathcal{N}(0, 1/\sqrt{256})$. We also compare our approach to the Monte Carlo estimation of dual kernel, i.e., $K_\sigma(x, y) \approx \frac{1}{m} \sum_{i=1}^{m} \sigma(\langle w_i, x \rangle)\sigma(\langle w_i, y \rangle)$ where $\{w_i\}_{i=1}^{m}$ are i.i.d. standard Gaussian vectors. In Figure 1, we plot relative errors of the Frobenius norm of kernel approximations in terms of wall-clock times (**top**) and polynomial degree (**bottom**). We run 10 independent trials and evaluate the average approxmation errors. We observe that our approximation with Hermite expansion outperforms the Monte Carlo method for all activations we used. In particular, sin and Gaussian are well approximated because they are smooth and norms of their derivatives are bounded with respect to the normal measure.

**Performance on CIFAR-10 classification.** We also benchmark the proposed CNTK approximating via sketching algorithm. We perform CIFAR-10 classification [57] by solving the ridge regression problem. The image classes are converted into 10-dimensional one-hot vectors and inputs are pre-processed with regularized ZCA [54, 58]. We report the best test accuracy among 20 choices of ridge parameters in $\{10^{-10+\frac{12}{19}i} \mid i = 0, 1, \ldots, 19\}$. We extract CNTK features of a 5-layer convolutional neural network (known as Myrtle5 [54]) without pool-

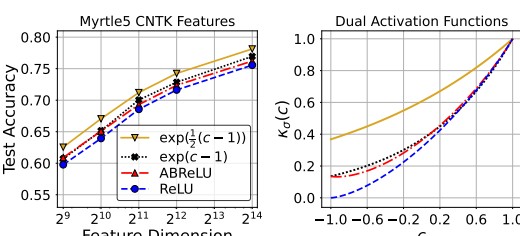

Figure 2: Test accuracy of CIFAR-10

ing by setting degree $q = 8$ and explore feature dimension $m = \{2^9, \ldots, 2^{14}\}$ and homogeneous dual kernels including ReLU, ABReLU activations as well as deep normalized Gaussian kernels with 2 scaling factors. See Appendix G for more details. In Figure 2, the test accuracy of neural kernels (**left**) and the corresponding their dual activations (**right**) are plotted. The dual activation of

ABReLU is very similar to the normalized Gaussian without scaling and their test performances are also comparable. We observe that the scaled normalized Gaussian shows the best performance which achieves 78.13% while the ReLU CNTK features [41] shows 75.56% with the same runtime. This is because the coefficients decay of the normalized Gaussian is faster than that of the ReLU, which leads to a lower approximation error of sketching algorithm. We also perform comparison among different activation functions in neural kernels in Appendix E.

**Speedup.** We observe that the exact CNTK of Myrtle-5 constructs a kernel matrix of size $60,000 \times 60,000$ and achieves 86-87% test accuracy. However, this requires approximately 151 GPU hours. Under the same setting, our CNTK features for the normalized Gaussian kernel take about 1.4 GPU hours, i.e. a $106\times$ speedup. If we use less training data to construct $20,000 \times 20,000$ kernel matrix, the accuracy is about 77% accuracy and the runtime is 16.8 GPU hours in which our approximation is still $12\times$ faster without loss of accuracy. We believe such acceleration through our methods open the door to using neural kernels in a wide range of research domains.

## 6 Discussion

In this work, we introduced methods to efficiently compute neural kernels for general activations. As activation functions play an important role in modern neural network architectures, we hope our work could empower researchers to explore properties of activations in a more principled way. We are excited with sketching method's compute efficiency by orders of magnitude on highly performant neural kernels to open up applications in dataset distillation [29] or uncertainty critical problems [59] such as autonomous driving, healthcare and science.

## Acknowledgements

Amir Zandieh was supported by the Swiss NSF grant No. P2ELP2_195140. Insu Han and Amin Karbasi acknowledge funding in direct support of this work from NSF (IIS-1845032), ONR (N00014-19-1-2406), and the AI Institute for Learning-Enabled Optimization at Scale (TILOS). We thank Timothy Nguyen and Jeffrey Pennington for discussions and feedback on the project.

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
