# OpenReview forum: "Fast Neural Kernel Embeddings for General Activations"
_NeurIPS.cc/2022/Conference — NeurIPS 2022 Accept_

### Official Review · Reviewer_4Kjx · 2022-06-23

**Rating:** 7
**Confidence:** 4
**Soundness:** 3 good
**Presentation:** 3 good
**Contribution:** 3 good

**Summary:**

The NTK and NNGP are powerful models for understanding and emulating the performance of neural networks in certain conditions. They are also worthy predictors in their own right, and allow one to easily predict with principled uncertainty. These models require computations involving a kernel or covariance function, which is a certain Gaussian integral representing the covariances between activations for two different inputs. Closed form expressions for kernels are known for many activations, each one having been derived by hand in previous works (see table 1).

The authors introduce a method for approximating NNGP kernels via a polynomial series (Theorem 1, 2). Furthermore, a certain expression in the NTK can be related to the NNGP via a derivative (Theorem 3), so the results readily extend to the NNTK. A random sketching method is also provided to approximate large kernel matrices of deep networks.

The paper is concluded with some experiments that show that the methods quickly (in terms of both wall-clock and number of series terms) and accurately approximate known NNGPs/NTKs. The authors also observe a 106 x speedup on a toy task compared with the exact kernel evaluation, and show reasonable performance.

**Questions:**

- In Theorem 2, the derivatives need to be absolutely continuous. However, the first derivative of the ReLU is not absolutely continuous (it is not even defined in a classical sense at the origin). So presumably the first half of Theorem 1 does not apply to the ReLU. How then does (12) hold?
- What is the condition required to swap the order of the derivative and the expectation in line 672? Differentiating under the integral is valid when...? The second derivative of the ELU is unbounded, and also does not satisfy your weaker condition (44). Therefore Theorem 3 cannot be applied to the ELU. I don't think this condition is actually required. Differentiation under the integral holds in the sense of generalised functions/distributions/Schwartz functions, as long as the resulting integral is finite. This means that the derivatives will involve Heaviside, Dirac, derivative of Dirac etc.
    - Looking at this again, your Theorem 3 is actually almost identical to part of Theorem 6 of reference [44] that you cite. Their expression for \lambda_3 is your expression for (14), up to a normalisation. Their proof is very similar to yours, but more appropriately uses a Stein's lemma for tempered distributions i.e. Schwartz functions to differentiate under the expectation. I think it is fine to keep your Theorem 3 as is, but acknowledge that this was independently studied in [44] with a much milder condition using Schwartz functions, albeit for a different purpose. Alternatively, cite or reproduce this more general theorem.

**Limitations:**

I do not envisage any *direct* societal complications from this work. I hope it is not a cop-out to say that evaluation of societal impact is not necessary for this work, because the contributions are completely task, problem, data and even to some degree model agnostic. Downstream works that use the techniques introduced here should consider these impacts.

**Strengths And Weaknesses:**

Strengths:
- The method is generally applicable, i.e. the conditions in Theorems 1 and 2 are sufficiently mild a wide range of activations of interest. However, I am confused about the condition in Theorem 2 (see question below).
- The authors cite a good amount of relevant literature, which is nice to see. The authors might also be interested in "Stationary Activations for Uncertainty Calibration in Deep Learning", which places the Matern kernel in the context of NNGP kernels.

Weaknesses:
- I did some random sampling of the proofs in the appendix. It is quite possible that I missed some errors. I have some concerns. See "Questions" box.
- I found moving from Definition 1 to equation (5) a bit unnatural. In fact, the original motivation for the NNGP is as either (a) an inner product in an infinitely wide hidden layer with iid zero mean Gaussian weights or (b) the covariance of the output of a network. This is equation (5). Now (5) only depends on a bivariate gaussian (u,v)=(w.x, w.y), hence implying definition 1. Going in this direction is more natural IMO. In contrast, currently the authors go in the other direction and state " By generalizing this observation to vectors in R^d", without any apparent motivation for the generalisation. Note also that it is not actually a generalisation but an iff. A random vector is Gaussian iff all linear combinations are Gaussian. So while it is in principle possible to go from definition 1 to (5) without "generalizing this observation to vectors in R^d", it is not really required to be this abstract. I suggest the authors start with (5) and move to definition 1. That being said, there is nothing technically wrong with the current presentation apart from the word "generalizing".


Typos:
- "NTK can be exactly computed using recursively" removing "using".
- Change "We believe such acceleration thru our methods would open the door to using neural kernels in a wide range of research domains." to "We believe such acceleration through our methods open the door to using neural kernels in a wide range of research domains."

---

> ### Author Response · Authors · 2022-08-02
> **Response to Reviewer 4Kjx**
>
> Thank you for the feedback and suggestions. Regarding the questions you raised:
>
> >In Theorem 2, the derivatives need to be absolutely continuous. However, the first derivative of the ReLU is not absolutely continuous (it is not even defined in a classical sense at the origin). So presumably the first half of Theorem 1 does not apply to the ReLU. How then does (12) hold?
>
> - Theorem 2 indeed contains 2 independent results; (1) error bound of truncated dual kernel approximation for absolutely continuous activations (Eq (11)) and (2) that for the ReLU activation (Eq (12)). We remark that Eq (12) is not the corollary of Eq (11). Our error analysis is based on the error bound of truncated Hermite expansion, and ReLU has a linear convergence rate (see Eq (40)) even if it is not absolutely continuous at the origin. More details can be found in Lines 637-643 in the supplement material.
>
> >What is the condition required to swap the order of the derivative and the expectation in line 672? Differentiating under the integral is valid when...? The second derivative of the ELU is unbounded, and also does not satisfy your weaker condition (44). Therefore Theorem 3 cannot be applied to the ELU. I don't think this condition is actually required. Differentiation under the integral holds in the sense of generalised functions/distributions/Schwartz functions, as long as the resulting integral is finite. This means that the derivatives will involve Heaviside, Dirac, derivative of Dirac etc.
>
> - We appreciate the reviewer pointing this out. As the reviewer mentioned, as long as the derivative of the dual activation (i.e., RHS in Eq (14)) is finite, exchanging integral with the partial derivative holds. In fact the condition of Theorem 3 can be relaxed to only requiring the integral of the second derivative of \sigma against the Gaussian measure being finite. This is sufficient for Eq(48) and Eq(51) to make sense. After this modification our Theorem 3 will be stronger than Theorem 6 of [44], which only applies to activation functions $\sigma(\cdot)$ that are bounded by polynomials.
>
>   We will make the condition in the proof of Theorem 3 more general so that it can cover the ELU activation and will properly cite [44] and mention that they first proved a similar result.
>
> > I found moving from Definition 1 to equation (5) a bit unnatural. In fact, the original motivation for the NNGP is as either (a) an inner product in an infinitely wide hidden layer with iid zero mean Gaussian weights or (b) the covariance of the output of a network. This is equation (5). Now (5) only depends on a bivariate gaussian (u,v)=(w.x, w.y), hence implying definition 1. Going in this direction is more natural IMO. In contrast, currently the authors go in the other direction and state " By generalizing this observation to vectors in R^d", without any apparent motivation for the generalisation. Note also that it is not actually a generalisation but an iff. A random vector is Gaussian iff all linear combinations are Gaussian. So while it is in principle possible to go from definition 1 to (5) without "generalizing this observation to vectors in R^d", it is not really required to be this abstract. I suggest the authors start with (5) and move to definition 1. That being said, there is nothing technically wrong with the current presentation apart from the word "generalizing".
>
> - We definitely agree that the reviewer’s suggestion sounds more natural. We will change definition 1 as well as all typos in our final version draft.

---

> > ### Comment · Reviewer_4Kjx · 2022-08-07
> > **response**
> >
> > Thanks for answering my questions. I am happy to recommend acceptance.

---

> > ### Comment · Reviewer_4Kjx · 2022-08-09
> > **caution**
> >
> > I just wanted to caution the authors when preparing their revision regarding the condition of [44] of the polynomial.
> >
> > > As the reviewer mentioned, as long as the derivative of the dual activation (i.e., RHS in Eq (14)) is finite, exchanging integral with the partial derivative holds. In fact the condition of Theorem 3 can be relaxed to only requiring the integral of the second derivative of \sigma against the Gaussian measure being finite. This is sufficient for Eq(48) and Eq(51) to make sense. After this modification our Theorem 3 will be stronger than Theorem 6 of [44], which only applies to activation functions  that are bounded by polynomials.
> >
> > As stated, this is not quite true. The polynomial condition in [44] is used because it is used to ensure that the integral does indeed act as the mapping computed by a tempered distribution. This makes use of the fact that the expectation of polynomials of a Gaussian are finite. The Gaussian pdf helps to ensure that the integrand involves a Schwartz function, but it is not always true if the activation is not bounded by a polynomial. There are probably other conditions you can use, but you will need *something*. To give a sense of when things could break, consider
> > $ \frac{\partial }{\partial t} \int_{-\infty}^\infty \Theta(x + t) e^x d x$. If differentiation under the integral were allowed, this would evaluate as $\int \delta(x+t) e^x dx = e^{-t}$. However, if you first try and evaluate the integral, you will see that it does not converge.
> >
> > The authors should worry about conditions on similar such conditions on $\sigma$, e.g. reciprocal activations $\sigma(x) = 1/x$  (perhaps it is easier to use the condition in [44] after all, if this covers all your cases of interest).

---

### Official Review · Reviewer_LAi7 · 2022-07-07

**Rating:** 7
**Confidence:** 3
**Soundness:** 3 good
**Presentation:** 3 good
**Contribution:** 3 good

**Summary:**


In this paper, the authors try to accelerate the computation of NTK with general activation functions. For the q-homogeneous dual kernel,  the authors extend the finite Hermite polynomials approximation technique (Daniely et al. 2016)  and provide approximation bounds. A sketching method is further proposed for the approximation of NTK.   The proposed sketching technique extends the technique (Zandieh et al.  2021) to homogeneous kernels (w.r.t more general activations) with rapidly convergent Taylor expansion.

**Questions:**

1. Does the approximation of NTK with general activation guarantee the global convergence property?

2. How to compute $L$-depth NTK recursively using Eq.(5) ? It is not clearly described in the context.

3. In the experiments,  what is the concrete procedure to compute the time in Figure 1?  It is better to compare with orthogonal random features (Yu et al. 2016) or other fast kernel approximation techniques and present the error vs. number of features besides the error vs. time.


Yu et al. Orthogonal Random Features. NeurIPS  2016.

Zandieh et al. Scaling Neural Tangent Kernels via Sketching and Random Features. NeurIPS 2021.

Daniely et al. Toward deeper understanding of neural networks: The power of initialization and a dual view on expressivity. NeurIPS 2016.

**Strengths And Weaknesses:**


Pros.
1. This paper focuses on NTK with general activation functions, which is an important direction for understanding deep neural networks with general activation functions.
2. A method (Theorem 3) is proposed to compute the dual kernel of the derivative of activations without knowing the activations.
3. The authors provide approximation bound of finite Hermite polynomials approximation to the q-homogeneous dual kernel.
4. The sketching approximation of NTK extends (Zandieh et al.  2021) to more general homogeneous kernels.
5. The paper is well organized and well written.



Cons.
1. An important property of NTK and other prior work with ReLU is the global convergence of training. This relies on the strictly positive definite property in NTK (with ReLU) or positive-definite of the Gram matrix. This may be part of the reason why ReLU is widely used in the assumption of prior work.   This paper focuses on the approximation of NTK with general activation functions; however, it may lose the global convergence property.
2. It seems that the fast sketching approximation technique (Sec.4) can only apply to $K_{\sigma}$ homogeneous dual kernel. The approximation in Eq.(15) is a straightfowrad Gauss-Hermite quadrature for each entry of the kernel $K$, which is brute-force and still with high complexity. There are still some steps needed before claiming to fill the gap for general activations.
3. In Figure 1,  how to compute the time? Is the sketching approximation technique more efficient than orthogonal random features or other fast kernel approximation techniques?
4. It is better to state more clearly the key technique challenging/improvement compared with (Zandieh et al. 2021).

---

> ### Author Response · Authors · 2022-08-02
> **Response to Reviewer LAi7**
>
> Thank you for the feedback and suggestions. Regarding the questions you raised:
>
> >Does the approximation of NTK with general activation guarantee the global convergence property?
> - In this paper, we consider the infinite-width limit where the NTK does not change during training. Unlike the finite-width setting, the kernel (ridge) regression problem with infinite-width NTK has a unique solution and can be directly solved. In the kernel regression setting, rather than convergence analysis, a generalization bound is often studied and it is known that a spectral approximation (as provided in Theorem 4) of the kernel matrix implies non-trivial upper bounds on the empirical risk (please see Lemma 4 in (Avron et al., 2017)).
>
>   For other kernel methods, a strict positivity of the smallest eigenvalue of NTK might be used for the convergence analysis. Similar to the NTK with ReLU activation, this property holds for non-polynomial activations when no two input data points are parallel. This can be readily followed by Section A.4.2 in (Gao et al., 2022).
>
>   \
>   Avron et al. "Random fourier features for kernel ridge regression: Approximation bounds and statistical guarantees." ICML, 2017.
>
>   Gao, Tianxiang, et al. "A global convergence theory for deep ReLU implicit networks via over-parameterization." ICLR 2022.
>
> > How to compute L-depth NTK recursively using Eq.(5) ? It is not clearly described in the context.
>
> - Given an analytic formula of dual activation $k_\sigma$ in Definition 1 and $K_\sigma^{(h-1)}(x,x),K_\sigma^{(h-1)}(x,y),K_\sigma^{(h-1)}(y,y)$  for some h, we can compute
>   - $K_\sigma^{(h)}(x,y) = k_\sigma\left(\sqrt{K_\sigma^{(h-1)}(x,x)}, \sqrt{K_\sigma^{(h-1)}(x,y)}, \frac{K_\sigma^{(h-1)}(x,y)}{\sqrt{(K_\sigma^{(h-1)}(x,x) K_\sigma^{(h-1)}(y,y)}}\right)$
>   - $K_\sigma^{(h)}(x,x) = k_\sigma\left(\sqrt{K_\sigma^{(h-1)}(x,x)}, \sqrt{K_\sigma^{(h-1)}(x,x)},1\right)$
>   - $K_\sigma^{(h)}(y,y) = k_\sigma\left(\sqrt{K_\sigma^{(h-1)}(y,y)}, \sqrt{K_\sigma^{(h-1)}(y,y)},1\right)$
>
>   and similar for ${\dot{K}_{\sigma}}^{(h)}$. This can be repeated for all h = 1,..., L and using Eq (4) we can obtain depth-L NTK and NNGP. We will clarify the details of recursive NTK computation in our draft.
>
> >In the experiments, what is the concrete procedure to compute the time in Figure 1? It is better to compare with orthogonal random features (Yu et al. 2016) or other fast kernel approximation techniques and present the error vs. number of features besides the error vs. time.
>
> - In Figure 1, we evaluate the wall-clock times for computing approximate 1000-by-1000 dual kernel matrices using the truncated Hermite expansion (Eq (9)). This is implemented in our supplementary material (please see lines 72-74, 88-90, 102-104 `ntk_activations_code/examples/dual_kernel_approx.py’).
>
>   Following the reviewer’s suggestion, we compare our method against ``orthogonal random features’’ (Yu et al., 2016) for the kernel approximation. We use an open-source python implmentation for generating orthgonal random vectors in https://github.com/neonnnnn/pyrfm/blob/master/pyrfm/random_feature/orthogonal_random_feature.py. For the case of ReLU activation, the number of features v.s. the relative kernel approximation error result is the following:
>
>   | Number of features | MC (Gaussian) | Orthogonal | Hermite (Our) |
>     |--------------------|---------------|------------|---------------|
>     |                512 |     9.163e-02 | 7.305e-02  |     6.919e-04 |
>     |               1024 |     6.397e-02 |  5.100e-02 |     6.657e-04 |
>     |               2048 |     4.582e-02 |  3.590e-02 |     3.683e-04 |
>     |               4096 |     3.234e-02 |  2.527e-02 |     3.846e-04 |
>     |               8192 |     2.300e-02 |  1.764e-02 |     2.401e-04 |
>     |              16384 |     1.628e-02 |  1.279e-02 |     2.565e-04 |
>     |              32768 |     1.148e-02 |  9.247e-03 |     1.743e-04 |
>     |              65536 |     8.107e-03 |  6.915e-03 |     1.858e-04 |
>
>   and other activations show the same tendency. Consequently, the orthogonal random features improve kernel approximation errors beyond the standard MCMC method, but still our Hermite expansion performs better. We will add more details and results of other activations in our final draft.

---

### Official Review · Reviewer_S3dK · 2022-07-12

**Rating:** 6
**Confidence:** 4
**Soundness:** 3 good
**Presentation:** 4 excellent
**Contribution:** 3 good

**Summary:**

The Neural Tangent Kenrel offers a compelling framework to (partially) understand some theoretical aspects of neural networks, especially near initialization. In the infinite-width regime, It has been known that NTK with ReLU activation has a closed-form analytic formula which enables exact computation of the NTK in this limit. However, the exact kernel computation for general activations is intractable. This paper proposes an approximate kernel computation method that leverages some tools from functional analysis (Hermite decomposition).


**Questions:**

- There is a similar NTK decomposition in the “Spherical Harmonics“ basis that appeared in [1] and [3]. Moreover, authors of [2] show a similar extension of the decomposition on S^d to the whole of R^d (Theorem 5 in [2]). Knowing that Spherical harmonics are linked to Legendre polynomials, it would be great if the authors could discuss similarities and differences with Hermite decomposition.
- A recent work ([4]) has showed that NTK computation (in finite-width setting) can be considerably accelerated with some computational tricks. Can the authors comment on the differences (a part from the finite versus infinite width settings of course). For instance, what is the difference between a kernel computed with [4] for width 100 and an equivalent NTK computed with the methods specified in this paper? What can you say about training/inference time?
- In the proof of Thm3, the condition on the second derivative can be further weakened to something like $|\sigma''(t)| \leq C_1 \exp(C_2 t^2 h(t))$ where $\lim_{t \to 0 }h(t) = 0$ and $h$ satisfies some mild conditions.


[1] Geifman et al (2020) “On the Similarity between the Laplace and Neural Tangent Kernels”

[2] Hayou et al. (2021) “Mean-field Behaviour of Neural Tangent Kernel for Deep Neural Networks”

[3] Hayou et al. (2021) “Stable ResNet”

[4] Novak et al. (2021) “Fast Finite Width Neural Tangent Kernel”

[5] Bietti 2021 “APPROXIMATION AND LEARNING WITH DEEP CONVOLUTIONAL MODELS: A KERNEL PERSPECTIVE”

**Limitations:**

Yes.

**Strengths And Weaknesses:**

**Strengths**:
- The paper is very well written and flows nicely. The problem is well introduced and the results are stated in a logical sequence which serves well the purpose of the paper.
- The theoretical results are sound and
- The proposed methods are theoretically supported and the numerical results show some superiority compared to traditional methods used for NTK computation.
- The proofs are well structured and easy to follow. I have one question about the proof of Thm3, see Questions section.

**Weaknesses**:
- I believe that it would benefit the paper if approximation rates are provided for general depth L networks instead of just depth 1 networks. The dependency on L would capture the growth of the approximation error as depth increases and maybe suggest when to use (or not use) one method or another. Besides, existing work such as [2] show that NTK tends to deteriorate with depth, I am keen to see what happens to the approximate NTK in this case.
- The paper lacks a more comprehensive literature review. I believe a more in depth discussion of existing results would immensely benefit the paper. See questions section.
- Numerical results are only reported for CIFAR10 (a part the synthetic dataset). It would be better if other (small, not necessarily ImageNet) datasets are included, this would help conclude wether activations such as the exponential or ABRelu is consistently better than ReLU. Also, results for varying depth would add value to the experiments section and make more complete.

---

> ### Author Response · Authors · 2022-08-02
> **Response to Reviewer S3dK**
>
> Thank you for the feedback and suggestions. Regarding the questions you raised:
>
> > I believe that it would benefit the paper if approximation rates are provided for general depth L networks instead of just depth 1 networks. The dependency on L would capture the growth of the approximation error as depth increases and maybe suggest when to use (or not use) one method or another. Besides, existing work such as [2] show that NTK tends to deteriorate with depth, I am keen to see what happens to the approximate NTK in this case.
>
> - In Theorem 4 (main draft), we indeed provide a ``spectral approximation’’ guarantee of depth-L NTK for homogenous activations. Our theorem implies that the feature dimension of NTK approximation increases when depth L becomes large in order to guarantee the constant spectral approximation error. For example, for the normalized Gaussian (Eq (13)), the feature dimension can depend on  poly$((\log n)^L)$ factor, where n is the number of data points.
>
>
> >There is a similar NTK decomposition in the “Spherical Harmonics“ basis that appeared in [1] and [3]. Moreover, authors of [2] show a similar extension of the decomposition on S^d to the whole of R^d (Theorem 5 in [2]). Knowing that Spherical harmonics are linked to Legendre polynomials, it would be great if the authors could discuss similarities and differences with Hermite decomposition.
>
> - Hermite series expansion allows us to transform the activation function to the dual kernel in Eq (5) when the random vector $w$ is sampled from the standard Gaussian distribution. When the random vector $w$ is drawn from the d-dimensional uniform sphere, one can make use of Legendre polynomials, similar to our approach. For the unit norm inputs, this is indeed the Mercer decomposition of dot-product kernels.
>
>
> >A recent work ([4]) has showed that NTK computation (in finite-width setting) can be considerably accelerated with some computational tricks. Can the authors comment on the differences (a part from the finite versus infinite width settings of course). For instance, what is the difference between a kernel computed with [4] for width 100 and an equivalent NTK computed with the methods specified in this paper? What can you say about training/inference time?
>
> - Thank you for pointing this out, we will add discussion in the next revision. [4] focuses on computing exact, finite-width NTK for given specific weights, while our work aims to approximate the infinite-width NTK (marginalized over all weights).
>
> - [4] could indeed be used for approximating the infinite-width NTK as well, but the authors do not perform any related analysis, and focus on other applications in section 2. Further, note that per Figure 2 (left) from [4], for CNNs a substantial benefit of [4] emerges only for the full-sized NTK of size $N O \times N O$ with $O = 1000$ output logits (at least on the GPU). In the infinite width, all $O$ output logits are i.i.d., and only the $N \times N$ kernel needs to be computed. Therefore we believe, at least for CNN kernels on GPUs, [4] could only provide a small improvement for the purpose of approximating the infinite-width NTK (unlike our work, demonstrating a 100x speed-up), and therefore the training time (time to construct the kernel) would be better with our method.
>
> - Regarding inference time, we remark that [4] does not allow to obtain NTK features (apart from the baseline method), since [4] achieves their speedup specifically by avoiding constructing features and computing the $N \times N$ NTK entries directly. Therefore the inference cost with their method is cubic _in the dataset size_ $N$, while inference with our low-rank kernel approximation takes time cubic _in the number of features_, which is usually much smaller than the dataset size $N$.
>
>
> >In the proof of Theorem3, the condition on the second derivative can be further weakened to something like $|\sigma''(t)| \le C_1 \exp(C_2 t^2 h(t))$ where $\lim_{t\rightarrow 0} h(t)=0$ and $h$ satisfies some mild conditions.
> - This is very interesting. We thank the reviewer for suggesting a more general condition for Theorem 3 as well as comprehensive literature. We will update them in the final version of our draft.

---

### Official Review · Reviewer_vFLm · 2022-07-16

**Rating:** 6
**Confidence:** 1
**Soundness:** 3 good
**Presentation:** 3 good
**Contribution:** 3 good

**Summary:**

This paper builds and extends the theory of infinite-width neural kernel computations. Compared with prior works, we can summarize its contribution as follows:
(1) It explicitly computes the NNGP and NTK kernels for a wider range of activation functions as shown in table 1.
(2) It derives an explicit expression of the dual kernel for polynomial activation, which also expends the assumption $x, y \in S^{d-1}$ to a more general one $x, y\in R^d$, and suggests using truncated Hermite expansion to approximate the dual kernel for non-polynomial activations.
(3) Given the situation that sometimes we are only given the dual kernel but not the corresponding activation or derivative of the activation functions, this paper propose to compute the dual kernel of the derivative without knowing the activation as shown in theorem 3.
(4) It uses random sketching techniques to accelerate NTK approximation, which is an extention to previous work which was only applicable for ReLU activation.

**Questions:**

I'm not familar with neural kernels before, so it's hard for me to provide some useful suggestions for this paper.

**Limitations:**

The limitations and potential negative social impact are not described since this is a theoretical work.

**Strengths And Weaknesses:**

Strength:

* This theoretical work extends the previous studies of dual kernels (NTK and NNGP), which is of high quality and might be useful as a good reference for future research in this area.

* Most theorems proved in this paper are incremental, but still preserves enough significance.

---

> ### Author Response · Authors · 2022-08-02
> **Response to Review vFLm**
>
> We thank the reviewer for the valuable comments.

---

### Meta-Review · Area_Chair_ykRr · 2022-08-29

**Recommendation:** Accept
**Confidence:** Certain

**Metareview:**

Most prior works on neural kernels have focused on using the ReLU activation. In this work, the authors provide new methods that can approximate multi-layered Neural Network Gaussian Process (NNGP) kernels and Neural Tangent Kernel (NTK) matrices for a wide range of activation functions. All the four reviewers recommended acceptance of the paper.

**Award:**

No

---

### Decision · Program_Chairs · 2022-09-14

Accept